# Policy and Guideline Review of Vaccine Safety for COVID-19 in Pregnant Women in Southern Africa, with a Particular Focus on South Africa

**DOI:** 10.3390/vaccines10122077

**Published:** 2022-12-05

**Authors:** Rujeko Samanthia Chimukuche, Busisiwe Nkosi, Janet Seeley

**Affiliations:** 1Africa Health Research Institute, KwaZulu-Natal, South Africa; 2Division of Infection & Immunity, University College London, London WC1E 6BT, UK; 3University of KwaZulu-Natal, KwaZulu-Natal, South Africa; 4Department of Global Health and Development, London School of Tropical Hygiene and Medicine, London WC1H 9SH, UK

**Keywords:** vaccination policies, policy implementation, COVID-19, pregnant and postpartum women, Southern Africa

## Abstract

Pregnant and lactating mothers have historically been excluded from clinical trials. To understand the shift from excluding to including this population in COVID-19 vaccine trials, we conducted a review of guidance issued by countries in southern Africa over the last three years. We conducted a review of documents and official statements recorded on Ministries of Health websites, and social media platforms, the World Health Organisation website, the COVID-19 Maternal Immunisation tracker and the African Union official webpage. Search terms included COVID-19 vaccination policies, guidelines for pregnant and lactating women, COVID-19 vaccination trials and pregnant women. We retrieved and reviewed policies, guidelines, and official statements from 12 countries. We found inconsistencies and incomplete guidance in respect to the inclusion of pregnant and lactating mothers in COVID-19 vaccine trials from the selected countries. Of the twelve countries reviewed, Namibia and South Africa had clear guidance on vaccination plans and implementation for pregnant women, and their inclusion in COVID-19 vaccine trials. Explicit and clear guidelines are critical in communicating changes in policy towards those deemed vulnerable for them to participate in vaccine trials. This review provides lessons for future pandemics on managing changes in guidance towards those groups historically excluded from vaccine and clinical trials.

## 1. Introduction

In 2020, clinical and vaccine trials for COVID-19 excluded pregnant and post-partum women [1]. More than 300 clinical trials investigated therapeutics for COVID-19 excluding pregnant women, despite many of these trials repurposing drugs already widely and safely, used in pregnancy [1]. The exclusion of pregnant and post-partum women from vaccine trials was because of concerns about the risk of harm to mother and baby due to concerns about vaccine safety [2,3]. As a result, the early 2021 COVID-19 randomized clinical treatments or vaccine trials did not focus on pregnant women therefore data on vaccine safety and immunogenicity have been limited [4,5,6]. Systematic reviews of COVID-19 clinical studies and registries conducted in 2021 revealed that 80% of clinical trials had ‘pregnancy’ as an exclusion requirement [7,8]. Studies have shown that maternal immunisation with select vaccines prevents illness to the pregnant mother and confers immunity to the infant [9], however, equally some vaccines, such as the Human Papillomavirus Vaccine and live influenza vaccine are not considered safe during pregnancy [10,11]. 

In this analysis we mapped COVID-19 vaccination policies or guidelines in sub-Saharan African countries (SSA) with a view to understand the shift towards inclusion after an initial period of exclusion of pregnant women and lactating mothers. Early COVID-19 clinical trials, for example, on the effectiveness of BNT162b2 vaccine against Omicron variant and the efficacy of the ChAdOx1 nCoV-19 vaccine against the B.1.351 variant in South Africa, excluded this population [12,13]. Focusing mainly on South Africa we assessed how policies were developed around this topic as compared to other countries in southern Africa. 

Protecting research participants from harm has traditionally been aligned with excluding certain populations from clinical trials. Pregnant women have been historically excluded from clinical and pharmacologic trials for several reasons including ethical concerns and possible harm caused by foetal exposure to the vaccine [14,15,16,17]. Therefore, this has led to insufficient data to make evidence-based recommendations, leaving pregnant women omitted from vaccine rollout plans and sometimes compromising their future health care [18]. During the early months of the COVID-19 pandemic, most vaccine companies took the precaution of excluding pregnant and lactating women from trials and access to the vaccine as a precautionary measure. This led to confusion and concern among pregnant and post-partum women about their safety during the pandemic and decision-making with limited data [14]. 

### Justification in the Inclusion of Pregnant and Postpartum Women in COVID-19 Vaccine Trials

As the pandemic progressed, the World Health Organisation acknowledged that pregant and post-partum women had a significantly higher risk of severe disease or death and should be prioritised in vaccine trials and rollouts [1,19]. Guidelines on the safety of COVID-19 vaccines were issued in April 2021 by the WHO Strategic Advisory Group of Experts on Immunization (SAGE) recommending that pregnant women receive COVID-19 vaccines if the benefits of vaccination outweigh the potential risks [20,21,22]. The guidelines stated that before offering COVID-19 vaccines to pregnant women, the women should be routinely informed about the benefits and anticipated potential and known risks of the vaccine as compared to the risks of the disease the vaccine is preventing. Evidence continued to build, showing that COVID-19 vaccination during pregnancy was safe and effective [23]. Nevertheless, some uncertainty remained in the absence of solid safety data provided by clinical trials [24].

There was a shift towards the inclusion of pregnant women in vaccine development and trials with countries developing policies and guidelines on how to use the vaccines based on WHO recommendations [23,25]. Initiatives to involve this population have been underway since then, with the first vaccine trial including pregnant and lactating mothers conducted in the United States of America in early 2021 [24,26,27]. In 2021/2022 several countries have included pregnant and lactating mothers in vaccine trials for example, on antibody increase in breastmilk from post-partum mothers who have been vaccinated against COVID-19 [28]. However, inconsistencies exist across countries, regarding policies on the inclusion of pregnant women in vaccine trials. 

## 2. Materials and Methods

### Design

To develop an understanding of how COVID-19 vaccination implementation has evolved over the last three years, we did a policy review of official statements, policies or guidelines. We analysed documents and official statements that were recorded on Ministries of Health websites and social media platforms pertaining to pregnant women and the safety of the COVID-19 vaccines in Southern Africa over the last three years (Table 1). Our key terms included and were not limited to: COVID-19 vaccination policies, guidelines for pregnant and lactating women, COVID-19 vaccination trials and pregnant women. RSC led the country-by-country internet search in Southern Africa. Documents were retrieved from official country and Ministries of Health websites, official social media pages, such Facebook, Twitter, World Health Organisation website, COVID-19 Maternal Immunisation tracker and the African Union official webpage. RSC and BN independently reviewed the documents. All the authors met regularly via Zoom and e-mail and reviewed selected documents and discussed the broad themes. 

We used a Microsoft Excel spreadsheet to record summarised data from the policies, including the title of the policy document, year of policy adoption, key message of the policy, and additional information regarding whether the policy content related to COVID-19 vaccination of pregnant women was explicit (stating clearly and in detail, providing details of how implementation should take place), or no explicit policy (there is some mention but no prescriptive detail to guide implementation although some guidelines are implemented) or no policy mentioned (no COVID-19 pregnant women policies or guidelines mentioned).

## 3. Results

A total of 12 southern Africa country policies, guidelines and official statements were reviewed by RSC and cross reviewed by BN. The countries included Angola, Botswana, Eswatini, Lesotho, Madagascar, Malawi, Mozambique, Namibia, South Africa, Tanzania, Zambia and Zimbabwe, Figure 1. Explicit policies were documented in South Africa, following the World Health Organisation guidelines and with two main themes emerging from the review which provided the focus for the data used in this paper: policy and guidelines on the safety of COVID-19 vaccines for pregnant and lactating mothers and COVID-19 vaccination campaigns for pregnant and lactating mothers.

### 3.1. Southern Africa Policy and Guidelines on COVID-19 Vaccination for Pregnant Women

Explicit policies and guidance statements were issued during 2021 for South Africa and Namibia, with clear guidance on vaccination plans and implementation for this population (Figure 1). Eight out of eleven countries analysed had no explicit policies, with little or no guidance on implementation of vaccination programs (Figure 1). However, official guidance statements on social media pages permitted vaccination and emphasised the safety and effectiveness for pregnant and lactating women. No recorded data were found for Lesotho and Madagascar, as illustrated in Figure 1.

### 3.2. Safety of COVID-19 Vaccines for Pregnant and Lactating Mothers

The review of the policy in South Africa showed that there was a shift in communication to include pregnant women in vaccinations in 2021, after the WHO statement in April 2021 that pregnant and post-partum women may receive the COVID-19 vaccine if the benefits of vaccination outweigh the potential risks [20,21]. 

The South Africa Health Products Regulatory Authority (SAHPRA), issued recommendations in April 2021 that considered the safety of the vaccines in pregnant and lactating women [29]. SAHPRA indicated pregnant and lactating women can be included in clinical trials in the early stages in vaccine development so that they are not excluded from subsequent use of vaccines. COVID-19 vaccination messaging guidelines were later released by the Technical Committee of the Inter-Ministerial Committee on Vaccinations in May 2021 to assist in public engagement activities and health promotion messaging on the vaccine rollout addressing key questions around vaccine administration [30]. The guidelines supported SAHPRA’s messaging, adding that vaccines that use the same viral vector as the single dose Ad26.COV2.S COVID-19 vaccine can be given to pregnant people in all trimesters of pregnancy [30,31].

The South Africa National Department of Health circulated official statements that emphasised the COVID-19 vaccination messaging concerning pregnant and lactating mothers stating that COVID-19 vaccination can be offered to women who are eligible to be vaccinated during lactation [32]. An updated circular was issued on the 29 August 2021 elaborating when and where pregnant and lactating mothers can be vaccinated, preferably during their antenatal and postnatal visits at their nearest health facility [33].

### 3.3. COVID-19 Vaccination Campaigns for Pregnant and Lactating Mothers

Official guidelines and communication stated that vaccination was to be administered free of charge based on the principles of Universal Health Coverage [30]. South Africa policy documents analysed revealed that pregnant women at a high risk of exposure to COVID-19, such as health workers or people who have comorbidities which add to their risk of severe COVID-19 disease, may be vaccinated in consultation with their health care provider [29]. Namibia on the other hand, issued the National Deployment and Vaccination Plan for COVID 19 vaccines in 2021 [34] and adopted the World Health Organisation Strategic Preparedness and Response Plan for the African Region, with the Namibian Ministry of Health and Social Services leading the programme on case management, infection prevention and control, risk communications, tracking the spread of COVID-19, and encouraging people including pregnant and lactating mothers to get vaccinated [35].

Communication in South Africa was spread widely on the benefits of COVID-19 vaccination and the higher risk of severe illness among pregnant women. Benefits of vaccination were also extended towards the protection of the baby with antibodies crossing through into the placenta. South Africa used the Ad26.COV2.S COVID-19 vaccine in vaccination campaigns and drives after the WHO recommendation for its use in pregnant and lactating women [29]. 

## 4. Discussion

Our findings show that many countries in the southern African region had national guidelines which lacked recommendations on COVID-19 vaccination among pregnant women. This was also observed in a study done by Nachega, 2022 [36], encouraging an update of national guidelines in African countries for stronger recommendation of COVID-19 vaccination for pregnant and lactating women based on efficacy data that was evident during the early pandemic. The shift from excluding pregnant women from vaccination in the early pandemic to including them later is not clear. Our analysis showed that efforts were made in South Africa after the WHO statement was given. Although not much vaccine efficacy data had been found earlier, through clinical trials, concerted communication was circulated via ministry of health and media channels.

Inconsistencies in policy responses and implementation have been observed during the COVID-19 pandemic, where some countries acted quickly to address the pandemic reacting to the immediate moment whilst others waited, considering implications for the future [37,38]. Even with no clear consensus or guidelines with regard to vaccination, there was sub-optimal implementation of vaccination programs amongst this population [39], and this was evident in the official statements on national social media pages, encouraging pregnant women and lactating mothers to vaccinate. The COVID-19 pandemic led to policy-practice gaps in health responses in times of crisis [40]. Lessons can be learnt from the COVID-19 pandemic on the importance of understanding the implementation of policies, taking into account the needs of the community and the vulnerable population [40,41]. 

Early exclusion of pregnant and post-partum women from early COVID-19 clinical trials led to their removal from essential health delivery programs [3]. This was experienced in earlier Ebola vaccine trials in 2015–2016, where insufficient data were available from the small number of exposures during pregnancy. This led to the exclusion of pregnant women from subsequent Ebola vaccine trials in 2018–2019 despite clear signals that Ebola-related diseases were worse during pregnancy and had deadly consequences for the woman and foetus [41,42]. The perception that pregnant or breastfeeding women are a “vulnerable population” needing protection from exploitative research studies has to a certain extent hindered progression of care. Ethically, pregnant and post-partum women have a right to evidence-based, scientifically proven health care. The development of guiding principles to allow testing of pregnant women in clinical trials is an imperative [42]. Health personnel involved in clinical research must be reminded about the importance of including pregnant women in their studies [43]. Important clinical end points must be defined when researching new vaccines in pregnancy, keeping in mind that they may be different from the nonpregnant population.

Our analysis was based on policy documents available on the internet which have the potential to bias the analysis. We may have excluded countries whose policy documents were not accessible on the internet. 

## 5. Conclusions

Our review shows inconsistencies in how Ministries of Health communicate polices and guidelines regarding the inclusion of pregnant women and lactating mothers in COVID-19 vaccines with respect to the countries reviewed. Explicit and clear guidelines are critical in communicating the shift to ensure that populations deemed vulnerable benefit from vaccine trials. These findings provide insights for future pandemics on the inclusion of populations historically excluded from vaccine and clinical trials. 

## Figures and Tables

**Figure 1 vaccines-10-02077-f001:**
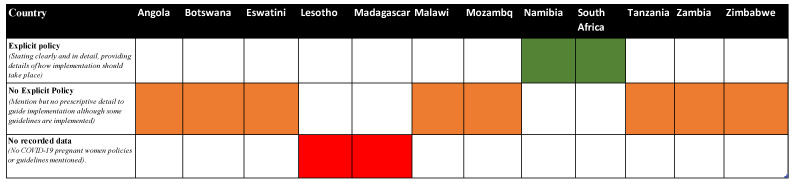
Indication of COVID-19 vaccination policies for pregnant women by country.

**Table 1 vaccines-10-02077-t001:** COVID-19 vaccination policies and official statements by country.

Country	Policy Guideline Source and Year	Type
South Africa	Guidance on the use of the Janssen Ad26.COV2.S (COVID-19) vaccine in pregnant and lactating women. South African Health Products Regulatory Authority (SAHPRA) “Vaccination of Pregnant and Breastfeeding Women” Director General Health. Republic of South Africa. 25 June 2021. VACCINATION OF PREGNANT AND BREASTFEEDING WOMEN.DIRECTOR GENERAL HEALTH. REPUBLIC OF SOUTH AFRICA. 29 August 2021 https://www.gcis.gov.za/vaccine-guideline (accessed on 19 October 2022).	Policies and guidelines
Angola	https://www.angop.ao/en/noticias/saude/minsa-preve-vacinar-10-mil-profissionais-do-pre-escolar/ (accessed on 3 October 2022) Official Portal of the Government of the Republic of Angola—News—NEW DECREE IN FORCE https://governo.gov.ao/ao/noticias/em-24-horas-2/ (accessed 27 July 2022)	Official statements on social media and government websites
Botswana	PUBLIC NOTICE ON THE ADMINISTRATION OF COVID-19 VACCINE SECOND DOSE.7 June 2021. https://web.facebook.com/OFFICIAL.MOHW.BW/photos/a.928167567330306/1955246084622444/?_rdc=1&_rdr (accessed 10 August 2022) https://web.facebook.com/OFFICIAL.MOHW.BW/photos/a.459580184189049/1972711179542601/?_rdc=1&_rdr (accessed 10 August 2022)	Official statements on social media and government websites
Eswatini	Good evening Eswatini. 2573 people...—Eswatini Government | Facebook. (accessed 27 July 2022)	Official statements on social media and government websites
Lesotho	No policy found on pregnant women and vaccines	
Madagascar	No policy found on pregnant women and vaccines	
Malawi	https://web.facebook.com/malawimoh/posts/306245058354441?_rdc=1&_rdr. (accessed 28 July 2022) https://www.nyasatimes.com/pregnant-women-can-now-get-covid-19-vaccine-ministry-of-health/ (accessed 28 July 2022)	Official statements on social media and government websites
Mozambique	https://covid19.ins.gov.mz/vacina-covid-19/ (accessed 24 July 2022)	Official statements on social media and government websites
Namibia	National deployment and vaccination plan for COVID 19 vaccines. Republic of Namibia. February 2021 https://m-partners.facebook.com/watch/?v=440512974037719&_rdr (accessed 22 September 2022) https://web.facebook.com/photo/?fbid=792232671449738&set=a.147401944250814&_rdc=1&_rdr (accessed 21 September 2022)	Policies and guidelines Official statements on social media and government websites
Tanzania	https://web.facebook.com/wizaraafyatz/posts/4491598857527213?_rdc=1&_rdr (accessed 21 September 2022)	Official statements on social media and government websites
Zambia	https://web.facebook.com/mohzambia/posts/statement-on-covid-19-in-zambialusaka-wednesday-12th-may-2021our-country-has-off/1906336979540950/?_rdc=1&_rdr (accessed 22 August 2022) https://web.facebook.com/mohzambia/photos/a.773733439467982/2076455569195756/?_rdc=1&_rdr (accessed 22 August 2022) https://web.facebook.com/mohzambia/photos/a.773733439467982/1997860320388615?_rdc=1&_rdr (accessed 22 August 2022	Official statements on social media and government websites
Zimbabwe	https://www.herald.co.zw/covid-19-vaccines-safe-for-pregnant-women/ (accessed 3 October 2022)	Official statements on social media and government websites

## Data Availability

Not applicable.

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
