# Peer review of "Policy and Guideline Review of Vaccine Safety for COVID-19 in Pregnant Women in Southern Africa, with a Particular Focus on South Africa"

_vaccines, 2022, doi:10.3390/vaccines10122077_

Round 1
Reviewer 1 Report
This short report by Chimukuche, RS, et al., addressed an important policy and guidline review of vaccine safety for COVID-19 in pregnant/lactating women in Southern Africa, with a particular focus on South Africa. The design of this work is very approachable, with a clearly compiled data and a good discussion.
Inclusion of pregnant and post-partum women in early COVID-19 vaccine clinical trial is very important, especially learning from the deadly consequences of excluding this particular population in 2015 Ebola vaccine trials in Africa. This report showed inconsistencies in how Southern Africa countries’ of Ministries of Health communicate policies and guidelines, and provide practical care and easy access of vaccination sites. Of the 12 countries included, South Africa is shown to be a good model for implementation of the processes involved.
The manuscript is well written, logical and linearly presented.
Author Response
Point 1:
This short report by Chimukuche, RS, et al., addressed an important policy and guidline review of vaccine safety for COVID-19 in pregnant/lactating women in Southern Africa, with a particular focus on South Africa. The design of this work is very approachable, with a clearly compiled data and a good discussion.
Inclusion of pregnant and post-partum women in early COVID-19 vaccine clinical trial is very important, especially learning from the deadly consequences of excluding this particular population in 2015 Ebola vaccine trials in Africa. This report showed inconsistencies in how Southern Africa countries’ of Ministries of Health communicate policies and guidelines, and provide practical care and easy access of vaccination sites. Of the 12 countries included, South Africa is shown to be a good model for implementation of the processes involved.
The manuscript is well written, logical and linearly presented.
Response 1
We thank you for the favourable comments
Reviewer 2 Report
Pregnant and lactating are generally excluded from clinical trials for ethical reason because of concerns about drugs toxicity for mothers and children. The principle of preventive measures applied is sometimes against the benefit for pregnant women particularly when the risk associated to the disease outweigh the toxicity risk of the new and innovative drug. As expected, clinical trials on COVID-19 vaccine at first excluded pregnant and lactating women. Evidences of worse outcomes in pregnant women with COVID-19 lead to shift in clinical trial practices toward inclusion of pregnant women. This paper describes inconsistencies existing across countries, regarding policies on the inclusion of pregnant women in vaccine trials.
1) The study was based on data available on internet that may induce information bias for countries with health policies information available online. This point need to be address at least in the discussion.
2) The procedure of reviewing the retrieved documents is not detailed. Who were the reviewers and were there cross review of the documents?
3) The materials used in this analysis are not clearly detailed by country. The documents used for each countries needed to be detailed in term of number and type for each country
4) The results focused on South Africa what about Namibia which seems to also have an explicit policy about vaccine in pregnant women? The comparison of the policies between South Africa and Namibia may be interesting
5) In describing the context, the authors should listed the COVID-19 clinical trials ongoing in the countries included in the analysis
Reviewer 3 Report
The brief report titled “Policy and Guideline Review of Vaccine Safety for COVID-19 in Pregnant Women in South Africa, with a Particular Focus on South Africa” discusses vaccine policy and guidance in the areas of Africa in the South. There was particular emphasis on whether there was guidance of any sort in each country’s health system dealing with vaccine use in pregnant and lactating women. It was noted that generally most pregnant and lactating women were excluded from original studies of the product. Overall, the authors looked for policies, guidelines, and official statements from 12 countries. In the introduction the authors describe the problem appropriately. And in a section labeled “Justification for the inclusion of pregnant and postpartum women in COVID-19 vaccine trials,” the authors discussed benefits, and at least mentioned the anticipated potential and known risk for vaccine adverse events as compared to risk of the disease the vaccine is preventing. Initially, there was no solid evidence about safety and effectiveness of vaccine in this very limited population.
In the materials and methods section, the authors reviewed documents from each country with particular emphasis on whether the policy for pregnant or lactating women was explicit, not explicit, or not mentioned at all in writing or media in the country's policies. Figure 1 notes the inconsistencies regarding policies and particularly the lack of detail even in places where policies did exist, with the exception being South Africa and Namibia where details were given.
This brief paper was not designed to look at outcome data of vaccine use, details regarding adverse events from the vaccine, and only briefly mentions the issue of healthcare workers. The paper, however, does make the point that with diseases that are significant and can be fatal, the issue of whether to include pregnant or lactating women in the initial trials is important. It does touch a bit on consent with risk versus benefit, the fact that policies are somewhat localized in their detail, and even occasionally, if there is no policy at all.
Despite the paper being nondetailed, descriptive, and not based on specific clinical outcome measures, it does shine a light on a problem that has existed in the past, still exists, and, unfortunately, is likely to continue to exist. That will be the primary value of the paper.
Author Response
Comments and Suggestions for Authors
Point 1: The brief report titled “Policy and Guideline Review of Vaccine Safety for COVID-19 in Pregnant Women in South Africa, with a Particular Focus on South Africa” discusses vaccine policy and guidance in the areas of Africa in the South. There was particular emphasis on whether there was guidance of any sort in each country’s health system dealing with vaccine use in pregnant and lactating women. It was noted that generally most pregnant and lactating women were excluded from original studies of the product. Overall, the authors looked for policies, guidelines, and official statements from 12 countries. In the introduction the authors describe the problem appropriately. And in a section labeled “Justification for the inclusion of pregnant and postpartum women in COVID-19 vaccine trials,” the authors discussed benefits, and at least mentioned the anticipated potential and known risk for vaccine adverse events as compared to risk of the disease the vaccine is preventing. Initially, there was no solid evidence about safety and effectiveness of vaccine in this very limited population.
In the materials and methods section, the authors reviewed documents from each country with particular emphasis on whether the policy for pregnant or lactating women was explicit, not explicit, or not mentioned at all in writing or media in the country's policies. Figure 1 notes the inconsistencies regarding policies and particularly the lack of detail even in places where policies did exist, with the exception being South Africa and Namibia where details were given.
This brief paper was not designed to look at outcome data of vaccine use, details regarding adverse events from the vaccine, and only briefly mentions the issue of healthcare workers. The paper, however, does make the point that with diseases that are significant and can be fatal, the issue of whether to include pregnant or lactating women in the initial trials is important. It does touch a bit on consent with risk versus benefit, the fact that policies are somewhat localized in their detail, and even occasionally, if there is no policy at all.
Despite the paper being nondetailed, descriptive, and not based on specific clinical outcome measures, it does shine a light on a problem that has existed in the past, still exists, and, unfortunately, is likely to continue to exist. That will be the primary value of the paper.
Response 1: We thank you for the favourable comments. We note the comments “Justification for the inclusion of pregnant and postpartum women in COVID-19 vaccine trials,” the authors discussed benefits, and at least mentioned the anticipated potential and known risk for vaccine adverse events as compared to risk of the disease the vaccine is preventing. Initially, there was no solid evidence about safety and effectiveness of vaccine in this very limited population.” We initially set out to determine what informed the shift (from exclusion to inclusion of pregnant and lactating mothers in vaccine), was it based on science, ethics, or political landscape, and we were not able to find the evidence. We then sought information on policy guidance and as the reviewer notes: “… with diseases that are significant and can be fatal, the issue of whether to include pregnant or lactating